# Correlates of Person-Specific Rates of Change in Sensor-Derived Physical Activity Metrics of Daily Living in the Rush Memory and Aging Project

**DOI:** 10.3390/s23084152

**Published:** 2023-04-21

**Authors:** Aron S. Buchman, Tianhao Wang, Shahram Oveisgharan, Andrea R. Zammit, Lei Yu, Peng Li, Kun Hu, Jeffrey M. Hausdorff, Andrew S. P. Lim, David A. Bennett

**Affiliations:** 1Rush Alzheimer’s Disease Center, Department of Neurological Sciences, Rush University Medical Center, Chicago, IL 60612, USA; 2Rush Alzheimer’s Disease Center, Department of Psychiatry and Behavioral Sciences, Rush University Medical Center, Chicago, IL 60612, USA; 3Medical Biodynamics Program, Division of Sleep and Circadian Disorders, Brigham and Women’s Hospital, Boston, MA 02115, USA; 4Division of Sleep Medicine, Harvard Medical School, Boston, MA 02115, USA; 5Rush Alzheimer’s Disease Center, Department of Orthopedic Surgery, Rush University Medical Center, Chicago, IL 60612, USA; 6Center for the Study of Movement, Cognition and Mobility, Neurological Institute, Tel Aviv Sourasky Medical Center, Tel Aviv 6492416, Israel; 7Sagol School of Neuroscience, Sackler Faculty of Medicine, Tel Aviv University, Tel Aviv 6997801, Israel; 8Department of Physical Therapy, Sackler Faculty of Medicine, Tel Aviv University, Tel Aviv 6997801, Israel; 9Division of Neurology, Department of Medicine, Sunnybrook Health Sciences Centre, University of Toronto, Toronto, ON M4N 3M5, Canada

**Keywords:** physical activity, aging, wearable sensors, linear mixed-effect model, multivariate modeling

## Abstract

This study characterized person-specific rates of change of total daily physical activity (TDPA) and identified correlates of this change. TDPA metrics were extracted from multiday wrist-sensor recordings from 1083 older adults (average age 81 years; 76% female). Thirty-two covariates were collected at baseline. A series of linear mixed-effect models were used to identify covariates independently associated with the level and annual rate of change of TDPA. Though, person-specific rates of change varied during a mean follow-up of 5 years, 1079 of 1083 showed declining TDPA. The average decline was 16%/year, with a 4% increased rate of decline for every 10 years of age older at baseline. Following variable selection using multivariate modeling with forward and then backward elimination, age, sex, education, and 3 of 27 non-demographic covariates including motor abilities, a fractal metric, and IADL disability remained significantly associated with declining TDPA accounting for 21% of its variance (9% non-demographic and 12% demographics covariates). These results show that declining TDPA occurs in many very old adults. Few covariates remained correlated with this decline and the majority of its variance remained unexplained. Further work is needed to elucidate the biology underlying TDPA and to identify other factors that account for its decline.

## 1. Introduction

Adults in their eighth and ninth decades are the fastest-growing segment of our aging population. Hence, interventions that can prevent or slow the growing burden of progressive cognitive and motor decline, a common occurrence in our aging populations, is a public health priority [1]. A more active lifestyle has been reported to be associated with independent living, better health, and longevity [2,3]. Aging research has, therefore, focused on physical activity intervention studies and identifying other modifiable factors to advance public health efforts to slow or prevent functional decline in older adults. 

Advances in technology now afford researchers the opportunity for continuous monitoring and quantification of physical activity during daily living via unobtrusive sensors. These sensors circumvent the limitations of traditional self-reported questionnaires and capture both exercise and habitual physical activity that may comprise a larger percentage of physical activity during daily living in aging adults [4]. Another advantage of these sensors is that a single multiday recording of everyday living can yield metrics of multiple behavioral phenotypes, i.e., daily activity, sleep fragmentation, and circadian rhythms that cannot be captured during a short-supervised testing session. Over the past decade due to ease of use, availability, and falling cost, wearable sensors have become standard for phenotyping physical activity in older adults in both observational and physical activity intervention studies [5,6,7]. Yet, despite their increased use, few studies have quantified the person-specific rates of change of total daily physical activity in relatively old adults (e.g., octogenarians) or identified the factors that may contribute to person-specific slope of decline [8,9,10].

Characterizing the person-specific trajectories of declining daily physical activity and identifying the correlates of the slopes of decline are crucial for advancing efforts to prevent its decline and facilitate a more active lifestyle in aging adults. For example, recent work in this cohort has shown that different combinations of brain pathologies may be associated with the person-specific rates of declining cognitive and motor function in the same individuals [11,12]. Studies have tried to assess change in daily physical activity [13,14]. In a previous study we used a modeling approach that provided average change for the entire cohort, but this study did not characterize person-specific rates of change, i.e., slope of longitudinal daily physical activity trajectories or its correlates [8]. Thus, despite the availability of diverse wearable sensors, there is a paucity of available data about person-specific rates of change of total daily physical activity and its correlates in aging adults. 

To fill the knowledge gaps about the person-specific rates of change of total daily physical activity and its correlates, the current study extracted total daily physical activity metrics from multiday recordings of everyday living derived from a wrist-worn accelerometer worn by more than 1000 community-dwelling older adults participating in the Rush Memory and Aging Project [15]. These individuals underwent a structured baseline exam that collected a wide range of covariates that may affect motor abilities and physical activity. This study employed linear mixed-effect models to identify the correlates of the level and person-specific slopes, i.e., changes in the annual rate of total daily physical activity in community-dwelling older adults. 

## 2. Materials and Methods

### 2.1. Participants

Participants were from the Rush Memory and Aging Project [15]. The study recruited residents of continuous care retirement communities, subsidized housing, and through local churches and social service agencies serving minorities and low-income elderly in metropolitan Chicago. Written informed consent was obtained, and the study was conducted in accordance with the latest version of the Declaration of Helsinki and was approved by Rush University Medical Center Institutional Review Board.

Rolling admission for the Memory and Aging Project began in 1997, and actigraphy data collection was added in August 2005. Thus, the analytic baseline for this study was defined as the first cycle during which a participant completed multiday sensor recordings. At the time of these analyses, of 2258 participants recruited since actigraphy was added, 20 had not completed baseline testing and 840 did not have actigraphy testing (died before actigraphy started or refused). Of 1398 who had completed sensor testing, 315 were excluded as they did not have a follow-up testing session to allow longitudinal analyses leaving 1083 individuals for the current analyses. 

### 2.2. Assessment of Total Daily Physical Activity

Total daily activity was measured with actigraphs. Actical^®^ (Philips Healthcare, Andover, MA, USA) is a portable, battery-operated activity monitor similar in size to a wristwatch that was worn on the non-dominant wrist for 24 h/day for up to ten consecutive days. Actical accelerometers generate a signal proportional to the magnitude and duration of the detected motion. After its signal is digitized, it is rectified, integrated across 15 s, and rounded to the nearest integer to create an “activity count” for each 15-s epoch during which activity was recorded [8]. There were a total of 5760 epochs during each complete day, i.e., 24-h recording. We calculated the sum of all the activity counts for all the epochs for each day that was recorded. Total daily physical activity used in these analyses was the average total daily activity counts for all complete days recorded during a single cycle. Day-to-day reliability of baseline actigraphy based on variance components analysis has been reported in prior publications [4]. Reports by other investigators have examined activity counts for several common performances including sitting and writing for 3 min (43 activity counts), floor sweeping for 3 min (1721 activity counts) and walking for 5 min at 2.5 m.p.h. (2355 activity counts) [5]. 

### 2.3. Other Covariates

Our prior work has shown that the diverse groups of clinical covariates included in these analyses were related to the level of total daily physical activity [16]. In these analyses, we extend that study to examine person-specific trajectories. Thirty-two covariates from seven different groups measured at the study baseline are summarized in Table 1.

#### 2.3.1. Demographics

*Age* at study baseline was computed from the self-reported date of birth and date of actigraphy collection. Sex, years of education, race, and income were recorded at the parent study entry [15]. 

#### 2.3.2. Sensor Metrics

These metrics were extracted from the same continuous multiday recordings of everyday living obtained via the wrist-worn accelerometers that were used to extract total daily physical activity metrics (Section 2.2). Sleep fragmentation (*Kra*) summarizes the degree of fragmentation of sustained rest/sleep [6,7,8,9,10]. Circadian rest–activity metrics include *IS* the inter-daily variability of activity and *IV* the intra-daily variability of [17]. Fractal metrics describe fluctuations of activity during the recording, i.e., temporal long-range correlations in motor activity fluctuations at two time-scale regions. We performed detrended fluctuation analysis of the fractal self-similarity of activity patterns across two times scales: Fractal alpha 1 ~1 min to 90 min and Fractal alpha 2 2 h up to 10 h [18,19,20]. The fractal measures examined in this study were derived from the same multiday activity recordings from which we extracted total daily physical activity metrics. From a theoretical perspective, total daily physical activity, and the fractal measures assess different aspects of movement: fractal measures describe the randomness of physical activity during different time scales that are theoretically independent of the magnitude of total daily physical activity [18,19,20,21,22,23].

#### 2.3.3. Motor Function

Motor abilities: Grip and pinch dynamometry measured strength. The number of pegs placed in the Purdue Pegboard and the rate of finger tapping were recorded. Gait speed and the number of steps taken during an eight-foot walk and a 360° turn were measured. Standing for ten seconds on each leg and then toe stands were documented. These measures were scaled and summarized as a global motor score as previously published [24,25]. 

Global parkinsonism score: A previously validated continuous measure of parkinsonism was based on assessment with a modified Unified Parkinson’s Disease Rating Scale (UPDRS) [26,27]. These items were used to create scores for four parkinsonian signs that were averaged to provide a continuous global parkinsonian score (0–100), with higher scores indicating increased severity [27].

#### 2.3.4. Self-Reported Activities

Physical activity was based on questions adapted from the 1985 National Health Interview Survey about hours of activity/week spent engaged in five activities [28]. The frequency of social activity was based on six items involving social interaction [29]. The frequency of cognitive activities was based on 7 items involving cognitive activities [30].

Disability was assessed annually via three self-report instruments [31]. Mobility disability was assessed using the Rosow-Breslau scale that assesses three activities involving movement [32]. Basic activities of daily living (ADLs) were assessed using a modified version of the Katz scale that assesses six daily activities [33]. Eight instrumental activities of daily living (IADLs) were adapted from the Duke Older Americans Resources and Services project [34]. Participants were classified as being disabled if they required help or were unable to perform one or more items on each scale. 

#### 2.3.5. Cognitive Function

Detailed neuropsychological assessment was administered at annual intervals with a battery of performance tests, as previously described [35]. For the present analyses, we summarized the 19 individual measures to quantify five cognitive abilities. The five cognitive abilities that we used in the current analyses included: episodic memory (7 tests), semantic memory (3 tests), working memory (3 tests), perceptual speed (4 tests), and visuospatial ability (2 tests). Raw scores on individual tests were converted to z scores, using the baseline mean and SD of the entire cohort, and the z scores of component tests were averaged to yield composite scores [36]. Previous publications contain further information about the individual tests and the derivation of the different cognitive abilities [36].

#### 2.3.6. Chronic Health Conditions

Body mass index (BMI) was calculated based on measured weight in kilograms and height in meters squared [37]. Seven self-reported chronic health conditions were detailed [31]. The number of 3 self-reported vascular risk factors (i.e., hypertension, diabetes mellitus, and smoking) and 4 self-reported vascular diseases (i.e., myocardial infarction, congestive heart failure, claudication, and stroke) were used in these analyses [38].

#### 2.3.7. Psychosocial Factors

Depressive symptoms were assessed with a 10-item Center for Epidemiologic Studies Depression (CES-D) scale [39]. Social network size was quantified based on the number of family and friends each participant had and the frequency of interaction with them [30]. Social loneliness was based on five items from a modified de Jong-Gierveld Loneliness Scale [40]. Neuroticism, was based on six items from the neuroticism scale of the revised NEO personality inventory [41]. Baseline purpose in life was based on a 10-item scale derived from Ryff’s Scales of Psychological Well-Being [42].

### 2.4. Statistical Analysis

An initial review of our data suggested that logarithm transformation was necessary to fit the total daily activity data by linear mixed effects models. For our initial models, we also added a quadratic term for non-linear change, but we found that the annual rate of change was best described by a linear term for time alone. So, we did not include a non-linear term in our modeling in these analyses. So, these analyses focus on the between-subject variation of the overall linear trends of the longitudinal trajectories of total daily physical activity. This variation can be fully captured by the linear mixed-effects model via the variance of the random slopes [43]. We used a series of linear mixed-effects models to examine the level and rate of change in repeated measures of total daily physical activity and their associations with age, sex, education, and other risk factors. The outcome for these models was repeated annual measures of total daily physical activity. We used a series of linear mixed-effects models to examine the level and rate of change of repeated measures of total daily physical activity and their associations with age, sex, education, and other risk factors. The outcome for these models was repeated annual measures of total daily physical activity. For individual *i*, let $TDPA_{ij}$ denote the total daily physical activity at visit *j*, $t_{ij}$ denote the time of measurement in years since baseline, and $risk_{i}$ denote a given risk factor. The basic model structure can be specified as below,
$$TDPA_{ij}=\alpha_{mean}+Age_{i}\times \alpha_{age}+Male_{i}\times \alpha_{male}+Educ_{i}\times \alpha_{educ}+risk_{i}\times \alpha_{risk}+\gamma_{i0}+\;t_{ij}\times \left(\beta_{mean}+Age_{i}\times \beta_{age}+Male_{i}\times \beta_{male}+Educ_{i}\times \beta_{edu}+risk_{i}\times \beta_{risk}+\gamma_{i1}\right)+e_{ij},$$
where $\alpha_{risk}$ and $\beta_{risk},$ respectively, denote the fixed effects of the risk factor on the baseline levels and the slope of change on TDPA, and $\gamma_{i0}$ and $\gamma_{i1},$ respectively, denote the person-specific random intercepts and slopes, which reflect the person-specific level of TDPA and the person-specific rate of change.

We began with an initial model with terms for age, sex, education, Time, and the interactions of Time with the demographic variables; Time is defined as the time (in years) since the analytical baseline. Our analysis proceeded in three stages. In the first stage, we examined each of the covariates alone adding a term for the covariate and its interaction with Time. The term for the covariate examined the association of the covariate with the baseline level of total daily physical activity. The term for the interaction of the covariate with Time examined the association of the covariate with the annual rate of change in the total daily physical activity. In separate models, we examined the associations of an additional 29 potential correlates from seven groups of covariates with the level and rate of change of total daily physical activity. All the models included terms for age, sex, education, and their interactions with Time. 

The second stage used forward selection to examine all of the covariates from each of the groups together. Adding a term for each covariate and its interaction with Time together in a single model would isolate the covariates from each group that showed the strongest correlation with the level and rate of change in total daily physical activity. As we have carried out in prior studies, to ensure that we did not exclude any potential significant association from stage 2, covariates were carried forward to the third stage if their *p*-values were smaller than 0.10 [44]. 

In a third stage that used backward elimination, we examined a model that included all the covariates that survived the second stage of our analysis together with age, sex, and education. We then repeated the last model including age, sex, education, and non-demographic terms that survived the third stage of analysis (*p* < 0.05) and calculated the percentage of the variance of declining total daily physical activity accounted for by demographics and non-demographic covariates as compared to an initial model that included only a term for Time. We also estimated the minimal and maximal variance explained by the non-demographic covariates that remained associated with the rate of change of total daily physical activity in the third stage of our analysis. All analyses were performed using SAS software version 9.4 (SAS Institute, Cary, NC, USA).

## 3. Results 

### 3.1. Baseline Description of the Analytic Cohort

There were 1083 adults included in these analyses. On average, total daily activity was measured for 9 days (9.6 days; SD = 1.21 days). Total daily physical activity counts/day ranged from 0.16 × 10^5^ counts/day to 13.56 × 10^5^ counts/day (mean: 2.71 × 10^5^ counts/day; SD = 1.55 × 10^5^ counts/day). The mean follow-up was 4.9 yrs. (SD = 3.05 yrs) and on average there were five repeated measures of up to ten days of total daily physical activity (median 4, IQR 2,7). The baseline characteristics of the analytic cohort for the covariates examined in this study are provided in Table 1. A heat map illustrates the inter-relationship of these baseline covariates and total daily physical activity that is shown in the top row (Figure 1).

### 3.2. Annual Rate of Change in Total Daily Physical Activity

A linear mixed-effect model was used to summarize both the level and annual rate of change of total daily physical activity with adjustment for demographic measures. This initial model included seven fixed-effects terms including time (annual rate of change in total daily activity), a term to assess the effect of age on the level of total daily activity: age (at study baseline), and a term age x time to examine the association of age with the annual rate of change in total daily physical activity. Terms for sex and education and their interactions with time were also included in this model.

On average total daily physical activity declined by 16%/year (Estimate = 0.163, S.E., 0.005, *p* < 0.001). Figure 2A shows, the crude paths of declining total daily physical activity (grey lines), and the average decline predicted by the model (bolded black line) are illustrated for a random selection of 100 individuals to highlight the heterogeneity of their decline. Inspection of the person-specific slopes showed that 1079 of 1083 (99.6%) had negative slopes and four had positive slopes.

Older age was associated with a faster rate of decline (Estimate = −0.004, S.E., 0.001, *p* < 0.001). Figure 2B illustrates that an individual 91 years old at baseline would show a 4% faster rate of declining total daily physical activity as compared to an individual 81 years old at baseline. In contrast an individual 71 years old at baseline would show a 4% slower rate of decline as compared to an individual 81 years old at baseline. On average men and women showed a similar rate of decline during follow-up (Estimate = 0.009, S.E., 0.010, *p* = 0.355). Each additional year of education was associated with a 0.3%/year slower rate of total daily physical activity decline (Estimate = 0.003, S.E., 0.001, *p* = 0.025).

### 3.3. Correlates of Annual Rate of Change in Total Daily Physical Activity

In the first stage of our analysis, we examined the association of each of the covariates shown in Table 1 with the level and annual rate of change in total daily physical activity in a separate linear mixed effect model that controlled for age, sex, and education. Fifteen of thirty-two (47%) measures were related to the annual rate of change of total daily physical activity; one to four measures from each of the seven groups (Table 2, Stage 1). 

In the second stage, we examined the covariates from each of the seven groups together to identify which measures from each group were most strongly associated with the rate of change in total daily physical activity (Table 2, Stage 2).

There were 11 non-demographic measures from the seven groups that remained associated with the rate of change in total daily physical activity. Age and education remained associated with declining total daily physical activity. Of the multiday sensor metrics, sleep fragmentation, interdaily stability, and the fractal alpha 2 remained associated with the rate of change of total daily physical activity. Motor abilities remained associated with declining total daily physical activity and attenuated the association of parkinsonian signs when included together in a single model. Of the self-reported activities, late-life cognitive activities and IADL disability were independently related to declining total daily physical activity. Of the five cognitive domains examined, episodic memory and perceptual speed remained associated with declining total daily physical activity. A summary measure of seven chronic medical conditions and BMI as well as purpose in life remained associated with declining total daily physical activity.

In the final third stage, we examined the 11 non-demographic covariates that survived stage two in a single model that also controlled for demographic measures, i.e., age, sex, and education. This final model showed that three non-demographic measures including motor abilities, fractal 2, and IADL disability remained associated with the annual rate of change in declining total daily physical activity (Table 3, Stage 3).

To assess the contribution of these covariates in explaining the variance of the person-specific decline of total daily physical activity, we examined them individually and together. We estimated the minimal and maximum percentage of the variance of the person-specific rates of decline of daily physical activity explained by each of the four non-demographic covariates. Motor abilities explained most of the variance of declining total daily physical activity, followed by IADL disability and fractal 2. Together these three non-demographic covariates accounted for about 9% of the variance of declining daily physical activity as compared to about 12% accounted for by age, sex, and education (Table 4).

Figure 3B contrasts the rate of declining total daily physical activity for two average female participants 81 years old and with 15 years of education with low and high levels of function of these three key covariates. While both showed declining total daily physical activity, the rate of decline for the individual with poor baseline function was about twice as fast compared to the women with good baseline function. Figure 3A emphasizes that nearly 80% of the variance of total daily physical activity remained unexplained despite the varied covariates interrogated in these analyses.

## 4. Discussion

This prospective observational study characterized the person-specific rates of change of quantitative metrics of total daily physical activity from more than 1000 community-dwelling older adults. While the decline was heterogeneous, all but 4 of 1083 participants showed negative slopes during an average follow-up of five years and their rates of decline were faster with increasing baseline age. A wide range of baseline covariates were correlated with declining daily physical activity when examined in separate models (Table 2). Yet, after variable selection, only age and three non-demographic covariates i.e., motor abilities, fractal metrics, and severity of IADL disability remained associated with the rate of declining daily physical activity (Table 3). Importantly, the majority of the Variance of declining daily physical activity remained unexplained since at best demographics and these three covariates together only accounted for about 20% of its variance (Table 4; Figure 3A). The ubiquitous decline of total daily physical activity documented in this study underscores the need for new approaches to determine which facets of this complex phenotype drive its decline. Further work elucidating the biology and factors underlying the correlates and the unexplained variance of declining total daily physical activity is crucial to inform on its prevention.

### 4.1. Novel Features of the Current Study

The current study extends our prior studies and work by others in several important ways. First, the primary outcome of longitudinal trajectories was derived from repeated measures of sensor metrics extracted from multiday recordings that captured both exercise and habitual physical activities during the everyday living of more than 1000 community-dwelling older adults. Second, the current analyses employed models that yielded person-specific estimates of the level and rate of change in total daily physical activity and not just the average decline for the entire cohort [8,9,10]. Third, the current study examined a diverse range of more than 30 potential correlates of total daily physical activity including both validated self-report instruments for late-life activities, disabilities, health conditions, and psychosocial factors as well as continuous performance based-measures of cognition and motor function (Table 1; Figure 1) [15]. Fourth, this study also examined several additional novel sensor metrics extracted from the same multiday recordings. These additional metrics include sleep fragmentation, and circadian and behavioral (fractal) rhythms, some of which may be amenable to modification [17,19,45].

### 4.2. Person-Specific Decline of Total Daily Physical Activity

The current study extends our prior smaller study that quantified the average rate of decline for the entire cohort and characterized the person-specific slopes for total daily physical activity [8]. Using this approach, we found that all but four of 1083 adults showed declining total daily activity over an average of five years of follow-up (Section 3.2). Of the demographic covariates examined, age showed the strongest correlation and was independently related to declining daily physical activity. Together age, sex, and education accounted for 12% of person-specific rates of decline in total daily physical activity. The negative interaction of age with the rate of change in total daily physical activity means that with increasing age there was an accelerated decline in total daily physical activity. Figure 2B illustrates this stark finding showing that the rate of daily physical activity decline for an individual 91 years old at baseline was 8% faster than a similar individual 71 years old.

The nearly universal decline of total daily physical activity and its acceleration with increasing age observed in the current study may reflect the accumulation of impairments in multiple physiologic systems crucial for the production of physical activity such as musculoskeletal, metabolic, cardiopulmonary, and central nervous systems. The strong associations of motor function and disability with declining daily physical activity are consistent with daily physical activity as a proxy for overall health as better underlying motor abilities would enable more physical activity. Yet, the pathologic basis for declining total daily physical activity is unclear. For example, prior work in this cohort has shown that common age-related neurodegenerative and many cerebrovascular disease pathologies are not related to levels of total daily physical activity in old adults [46]. Recent work suggests that several years prior to death, older adults undergo a marked acceleration of functional decline that may account for some but not all of the decline documented in this study [47]. Further work is needed to elucidate the diverse mechanisms that drive declining physical activity to isolate mechanisms amenable to modification and treatment.

Prior work in this cohort has shown that total daily physical activity predicts varied adverse health outcomes [48,49,50,51,52,53]. However, the sensor employed in this study does not capture many important facets of physical activity that may be modifiable. For example, the devices used in this study and the metrics extracted do not specify which specific activities or the duration of activity drive our findings. The study did not examine the intensity of activities or energy expenditure that are also important features of physical activity [41]. Further studies are needed to determine if our findings of nearly universal decline are observed for these other facets of physical activity and different types of activities. Extraction of multiday mobility metrics from these same wrist sensor recordings may add important data that can inform these knowledge gaps [54].

### 4.3. Correlates of Declining of Total Daily Physical Activity

Nearly half of the covariates examined in the current study were associated with the slopes of declining daily physical activity when examined alone (Table 2) but only three non-demographic measures including motor abilities, disability, and fractal measures survived selection modeling when diverse groups of covariates were examined together in the same model (Table 3). The specific motor measure employed in this study was a composite measure summarizing ten motor performances. Further work is needed to determine which specific motor performances are most strongly related to declining total daily physical activity or whether others provide more specificity.

The fractal alpha 2 measure showed strong associations with declining daily physical activity. Unlike other rhythmic metrics that are focused on oscillations/fluctuations at discrete time scales, fractal regulation is focused on the interrelationship between activity oscillations/fluctuations at different time scales (2–10 h), thus representing a temporal organization that is different from that of rhythmicity. When similar fractal analyses have been applied to other physiologic outcomes (e.g., cardiovascular or respiratory function, gait), the fractal measures have been shown to reflect the flexibility of the underlying systems, to adapt to constraints and challenges, and to the complexity and richness of the system [55]. In prior studies, the fractal measures have been related to heart rate, respiration, and gait and predicted adverse health outcomes [56,57,58,59,60,61,62,63,64]. Similarly, fractal alpha 2 has been shown to predict varied adverse health outcomes in older adults in this cohort [18,20].

While the exact mechanism underlying the fractal patterns in motor activity needs further elucidation, prior animal and human studies have made significant progress by demonstrating that endogenous circadian regulation may play a role in the generation and maintenance of fractal patterns. It may serve as a better measure of the circadian function in an ambulatory setting since it is robust despite behavioral and environmental changes that may mask other rhythmicity metrics [65,66,67]. Additionally, the fractal scaling may capture features of environmental complexity and richness that stimulate neural control systems regulating the levels of physical activity during the varied settings of daily living. Thus, while the exact mechanism needs further elucidation, the fractal measure may have potential as a robust biomarker for the future decline of daily physical activity.

Our findings that better motor function and less disability are associated with a slower decline in total daily physical activity suggest the possibility that there are additional behavioral or biological factors that may provide “resilience” that can offset age-related declining total daily activity (Table 3) [68]. The mechanisms underlying resilience are just beginning to be illuminated. Genes and proteins that may contribute to motor and cognitive resilience may be found in central nervous system tissues that contain neural control systems underlying these behaviors [68,69]. When we interrogated genes and proteins in the dorsal lateral prefrontal cortex, we discovered several genes and proteins associated with cognitive and motor resilience by regressing out the effects of measured brain pathologies [25,70,71,72,73]. These and other similar studies suggest that interrogating transcriptome or proteome may identify molecular mechanisms that lack a “pathologic footprint” and may either slow or hasten the decline of important aging phenotypes such as total daily physical activity. These genes and proteins may serve as novel therapeutic targets for drug discovery or for mechanistic studies to elucidate the biological mechanisms driving changes in late-life daily physical activity.

An unexpected but important finding in this study was that about 80% of the variance of declining total daily physical activity was unexplained (Figure 3A). This highlights the difficulty in isolating independently associated factors that improve risk prediction above conventional demographics for complex multifactorial aging phenotypes [74]. Yet, our finding that the majority of the variance of declining daily physical activity remained unexplained despite examining a wide variety of behavioral, physiologic, and health covariates underscores knowledge gaps not only about the trajectories of declining total daily physical activity but also about its correlates and underlying biologic substrate.

This study has several limitations. The current findings will need to be replicated in more diverse populations. The analyzed data are from an observational study, so the correlations observed with total daily physical activity may not be causal but rather reflect a shared underlying biology leading to the loss of diverse cognitive and physical functions. However, due to the select nature of this cohort, i.e., predominantly whites of European ancestry and higher than average education, it is likely that our findings may underestimate the rate of declining daily physical activity in the general population. Moreover, due to floor effects and co-linearity, this study could not model sedentary time and total daily physical activity. Nonetheless, multiday metrics from everyday living were obtained from a large number of well-characterized men and women, and the correlates of total daily physical activity were assessed using a wide range of previously published and validated instruments for both self-reported, performance-based, and sensor-derived covariates.

## 5. Conclusions

Nearly all the older adults in this study manifested negative slopes of daily physical activity that accelerated with increasing age. The idea that the person-specific slopes of total daily physical activity can be viewed as a proxy for overall health, calls attention to the added difficulties of intervention efforts to catalyze a more active lifestyle in aging adults. The frequent occurrence of cognitive and physical limitations in older adults adds to the costs that older adults must overcome to engage in a more active lifestyle. Decreasing function and poor health in older adults is an age-related barrier that may add to the individual barriers that are known to impede efforts facilitating an active lifestyle earlier in life. This also emphasizes the difficulties in generalizing solutions that are effective in younger to older adults. Due to the progressive loss of cognitive and motor function and the varied physical challenges faced by older adults, more varied and targeted strategies for improving physical activity among older adults will be needed. For example, efforts to increase habitual physical activity rather than formal exercise in older adults may be easier and less costly to implement in older adults.

Despite testing a very broad range of covariates, the majority of the variance of declining daily physical activity remained unexplained, underscoring the need for further studies to characterize the person-specific slopes of declining total daily physical activity in very old adults and to elucidate its unexplained variance. Future studies should also address standardizing how wrist-worn devices are used, how recordings are analyzed, and meaningful functional change to facilitate comparisons of studies conducted in different settings and populations. Many knowledge gaps impede public health efforts to modify the biology underlying declining total daily physical activity in our aging population or determine which facets of its decline might be amenable to modification. Ultimately, the usefulness of suggestions derived from observational studies, such as the current study, needs to be tested in randomized controlled clinical trials. However, it is not often considered that because of the wide-ranging forms of physical activity and its pervasive presence in many types of late-life activities during everyday living that it may not be possible to resolve the potential benefits of physical activity through randomized controlled clinical trials of older adults.

## Figures and Tables

**Figure 1 sensors-23-04152-f001:**
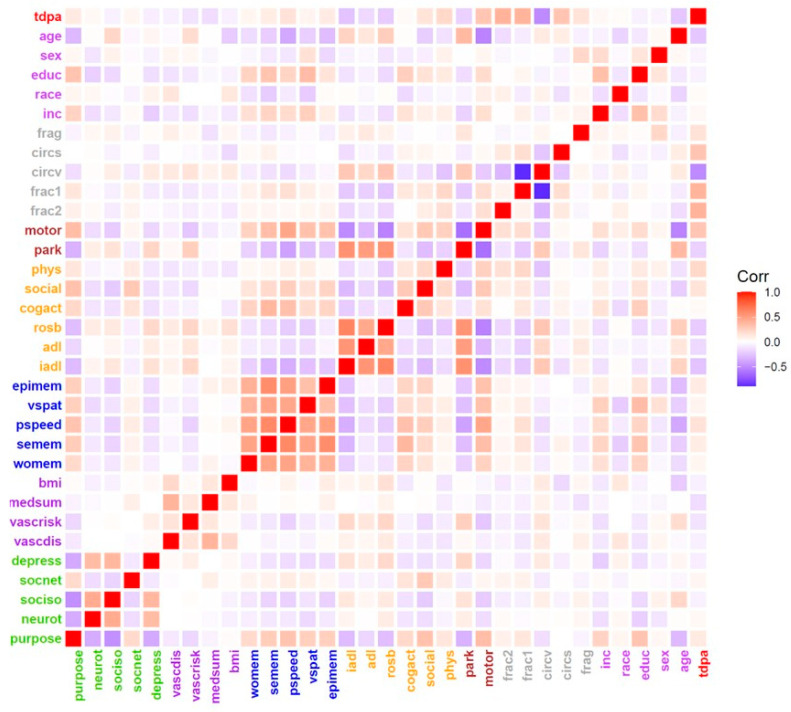
Baseline Correlations of Total Daily Physical Activity (TDPA) and Covariates *. *** Covariate Labels: (1) Demographics: age—**age; **sex—**sex; **educ—**education; **race—**race; **inc—**income; **(2) Sensor Metrics: frag—**sleep fragmentation; **circs—**circadian interdaily stability; **circv—**circadian interdaily variability; **frac1—**fractal alpha 1; **frac2—**fractal alpha 2; **(3) Motor Function: motor -**motor abilities; **park—**parkinsonism; **(4) Self-Report Activities: phys—**Physical activities; **social—**social activities; **cogact—**cognitive activities; **rosb**—mobility disability; **ADL—**activities of daily living; **IADL—**instrumental activities of daily living; **(5) Cognition: epimem—**episodic memory; **vspat—**visuospatial abilities; **pspeed—**perceptual speed; **semem—**semantic memory; **womem—**working memory; **(6) Chronic Health Conditions: BMI—**body mass index; **medsum—**chronic conditions; **vascrisk—**vascular risk factors; **vascdis—**vascular diseases; **(7) Psychosocial**: **depress—**depressive symptom; **socnet—**social network; **sociso—**social isolation; **neurot—**neuroticism; **purpose—**purpose in life.

**Figure 2 sensors-23-04152-f002:**
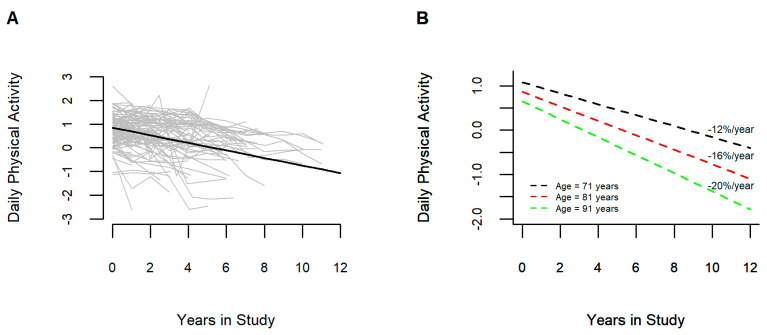
Trajectories of Declining Total Daily Physical Activity During this Study. (**A**) shows the crude paths of declining total daily physical activity (gray lines) and the average decline (solid black line) predicted by the linear mixed-effects model for a random sample of individual included in these analyses. (**B**) illustrates model derived trajectories of total daily physical activity for three average female participants with 15 years of education with different ages at baseline. Person-specific slopes of the rate of declining total daily physical activity for each individual are noted.

**Figure 3 sensors-23-04152-f003:**
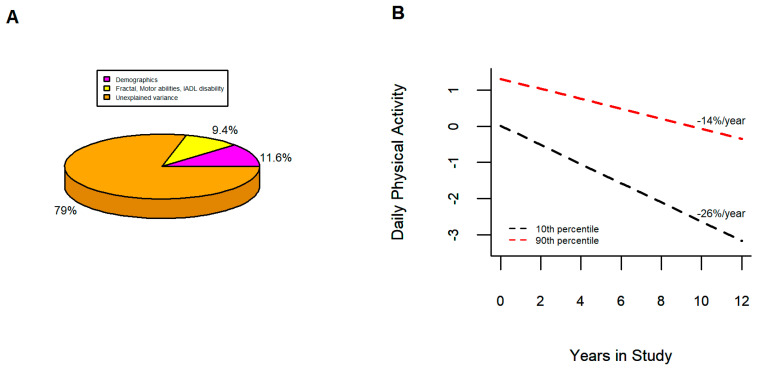
Trajectories of Total Daily Physical Activity in Adults with Low and High Levels of Key Clinical Covariates. (**A**) shows that demographic and three key covariates explained only about 20% of declining total daily physical activity. (**B**) illustrates model derived trajectories of two average participants, female 81 years old with 15 years of education, with low (black) versus high (red) function of key non-demographic covariates associated with the rate of change of total daily physical activity (10th vs. 90th percentile for continuous measures and high IADL disability score versus no disability. Both individuals showed declining daily physical activity, but the rate of decline was about twice as fast in the women with poor baseline function of key clinical covariates compared to the women with good baseline function.

**Table 1 sensors-23-04152-t001:** Baseline characteristics of the analytic cohort (N = 1083) *.

Covariate Groups	Variable at Baseline	Mean (SD) Median (IQR)
**Demographics**	Age (years)	80.9 (7.31)
	Female sex	76%
	Education (years)	15.1 (3.00)
	Race	5%
	Income (median level)	8 (5, 9)
**Sensor metrics**	Sleep fragmentation (0–1)	0.03 (0.01)
	Interdaily stability (0–1)	0.52 (0.12)
	Intradaily variability (0–2)	0.71 (0.19)
	Fractal alpha1	0.92 (0.06)
	Fractal alpha2	0.82 (0.10)
**Motor function**	Motor abilities (scaled)	1.02 (0.23)
	Parkinsonism (0–100)	7.00 (7.02)
**Self-report**	Physical activities (h/week)	2.7 (1, 4.8)
	Social activities (0–5)	2.7 (2.2, 3.0)
	Cognitive activities (0–5)	3.1 (2.9, 3.6)
	Mobility disability (0–3)	0 (0, 1)
	ADL disability (0–6)	0 (0, 0)
	IADL disability (0–8)	0 (0, 1)
**Cognitive function**	Episodic memory (z-score)	0.16 (0.74)
	Visuospatial abilities (z-score)	0.08 (0.84)
	Perceptual speed (z-score)	0.05 (0.80)
	Semantic memory (z-score)	0.13 (0.66)
	Working memory (z-score)	0.03 (0.74)
**Chronic health**	Body mass index	27.3 (5.39)
	Chronic medical conditions (0–7)	1.0 (0, 2)
	Vascular risk factors (0–3)	1 (1, 2)
	Vascular diseases (0–4)	0 (0, 1)
**Psychosocial factors**	Depressive symptoms (0–10)	0 (0, 1)
	Social network (0–9)	6 (3, 9)
	Social isolation (1–5)	2.0 (2.0, 2.4)
	Neuroticism (0–12)	7 (5, 9)
	Purpose in life average score (1–5)	3.7 (3.4, 4.0)

*** IQR interquartile range; Income**: ten levels $0–$4999 to $75,000 and over; median level was 8, $35,000–$49,999; **Sleep fragmentation**: higher values indicate more fragmented sleep; **Interdaily stability**: day to day similarity of the rest-activity pattern with higher values indicative of more stability; **Interdaily variability**: higher measure indicative of greater hour to hour fragmentation; **Fractal alpha 1 and 2** fractal correlations at lower (1.25–90 min and higher (2–10 h) time scales; lower values represent more random fluctuation patterns; **Social activities** average frequency of participation in 6 activities with higher values indicative of more frequent activities **Cognitive activities** average frequency of participation in 7 activities with higher values indicative of more frequent activities; **Social network:** higher scores indicative of more family or friends and more frequent interactions; **Social isolation:** higher scores indicative of feeling more isolated; **Purpose in life:** average score for ten items; higher score greater purpose.

**Table 2 sensors-23-04152-t002:** Selection of Key Covariates Associated with Declining Total Daily Physical Activity.

Group	Baseline Covariate × Time	Stage 1 Covariates Alone Estimate (S.E., *p*-Value)	Stage 2 Groups Covariates Together Estimate (S.E., *p*-Value)
**Demographics**	lag × age	−0.004 (0.001, <0.001)	**−0.004 (0.001, <0.001)**
	lag × sex	0.006 (0.010, 0.529)	0.006 (0.010, 0.567)
	lag × education	0.003 (0.001, 0.038)	**0.003 (0.002, 0.079)**
	lag × race	−0.005 (0.020, 0.782)	−0.004 (0.020, 0.825)
	lag × income	0.001 (0.002, 0.641)	0.001 (0.002, 0.666)
**Sensor metrics**	lag × sleep fragmentation	−0.899 (0.621, 0.148)	**−1.049 (0.62, 0.095)**
	lag × interdaily stability	0.041 (0.035, 0.243)	**0.065 (0.037, 0.078)**
	lag × interdaily variability	0.049 (0.023, 0.031)	−0.011 (0.056, 0.839)
	lag × alpha1	−0.176 (0.075, 0.019)	−0.225 (0.174, 0.198)
	lag × alpha2	−0.097 (0.046, 0.035)	**−0.107 (0.052, 0.040)**
**Motor function**	lag × motor abilities	0.092 (0.022, <0.001)	**0.077 (0.026, 0.003)**
	lag × global parkinsonism score	−0.003 (0.001, 0.002)	−0.001 (0.001, 0.284)
**Self-report activities**	lag × physical activities	0.001 (0.001, 0.587)	−0.001 (0.001, 0.571)
	lag × social activities	0.013 (0.008, 0.089)	−0.001 (0.008, 0.922)
	lag × cognitive activities	0.025 (0.007, <0.001)	**0.021 (0.007, 0.004)**
	lag × mobility disability	−0.014 (0.005, 0.002)	−0.005 (0.006, 0.349)
	lag × ADL disability	−0.012 (0.007, 0.101)	0.006 (0.008, 0.452)
	lag × IADL disability	−0.016 (0.003, <0.001)	**−0.014 (0.004, 0.002)**
**Cognition**	lag × episodic memory	0.024 (0.007, <0.001)	**0.017 (0.008, 0.028)**
	lag × visuospatial ability	0.006 (0.006, 0.271)	−0.005 (0.006, 0.447)
	lag × perceptual speed	0.021 (0.006, <0.001)	**0.014 (0.008, 0.071)**
	lag × semantic memory	0.022 (0.008, 0.004)	0.004 (0.010, 0.692)
	lag × working memory	0.016 (0.006, 0.009)	0.006 (0.007, 0.415)
**Chronic health**	lag × BMI	0.002 (0.001, 0.009)	**−0.002 (0.001, 0.013)**
	lag × chronic medical	−0.011 (0.004, 0.011)	**−0.012 (0.005, 0.014)**
	lag × vascular diseases	0.004 (0.007, 0.601)	0.006 (0.007, 0.388)
	lag × vascular risk factors	−0.004 (0.005, 0.489)	0.005 (0.006, 0.392)
**Psychosocial factors**	lag × depressive symptoms	0.000 (0.003, 0.987)	0.001 (0.003, 0.742)
	lag × social network	0.000 (0.001, 0.897)	−0.000 (0.001, 0.945)
	lag × social isolation	0.001 (0.007, 0.909)	0.007 (0.009, 0.444)
	lag × neuroticism	0.000 (0.001, 0.754)	0.001 (0.001, 0.5529)
	lag × purpose in life	0.021 (0.010, 0.044)	**0.029 (0.012, 0.016)**

This table shows the results of the first two stages of our analyses to select the covariates most strongly related to the rate of change in total daily physical activity. Each cell in the third column shows results (Estimate, S.E. [Standard Error] and *p*–value) from a single linear mixed effect model showing the association of a single covariate identified in the second column with the person-specific rate of change in total daily physical activity. Each model included an additional eight terms not shown including a term for time, baseline level of the covariate and terms for baseline age, sex, education, and their interaction with the rate of change of total daily physical activity. The second stage of our analyses used similar models that included all of the covariates from each group together in a single model to help select the covariates with the strongest associations with the rate of change in total daily physical activity from each of the seven groups. To ensure that we did not exclude potentially important covariates for the third stage of our analyses, we selected all covariates with a *p*-value of *p* < 0.10 (bolded).

**Table 3 sensors-23-04152-t003:** Identifying Covariates Independently Associated with Declining Total Daily Physical Activity.

Model Term	Stage 3 Final Model		Only Independent Covariates + Demographics	
	Estimate	S.E., *p*-Value	Estimate	S.E., *p*-Value
**Time**	−0.052	0.077, 0.495	−0.129	0.047, 0.006
**Age**	−0.008	0.003, 0.008	−0.006	0.003, 0.031
**Sex**	−0.064	0.041, 0.117	−0.059	0.041, 0.152
**Education**	−0.019	0.006, 0.002	−0.021	0.006, <0.001
**Episodic memory**	−0.006	0.027, 0.814		
**Perceptual speed**	0.048	0.027, 0.081		
**Sleep fragmentation**	15.013	2.524, <0.001		
**Interdaily stability**	1.168	0.138, <0.001		
**Fractal alpha2**	1.641	0.176, <0.001	1.746	0.186, <0.001
**Motor abilities**	0.316	0.101, <0.002	0.413	0.103, <0.001
**Late life cognitive activities**	−0.011	0.029, 0.699		
**IADL disability**	−0.052	0.016, 0.001	−0.074	0.017, <0.001
**BMI**	−0.000	0.003, 0.998		
**Medical conditions**	−0.069	0.017, 0.0000		
**Purpose in life**	0.007	0.040, 0.867		
**Lag** **× Age**	−0.002	0.001, 0.002	−0.002	0.001, 0.008
**Lag** **× Sex**	0.004	0.011, 0.698	−0.004	0.010, 0.726
**Lag** **× Education**	0.001	0.002, 0.530	0.002	0.001, 0.111
**Lag** **× Episodic memory**	0.013	0.007, 0.071		
**Lag** **× Perceptual speed**	−0.003	0.007, 0.720		
**Lag** **× Sleep fragmentation**	−0.893	0.612, 0.145		
**Lag** **× Interdaily stability**	−0.034	0.036, 0.338		
**Lag** **× Fractal alpha2**	−0.123	0.046, 0.008	−0.123	0.046, 0.007
**Lag** **× Motor abilities**	0.063	0.026, 0.014	0.084	0.024, <0.001
**Lag** **× Late life cognitive activities**	0.011	0.008, 0.164		
**Lag** **× IADL disability**	−0.013	0.005, 0.008	−0.013	0.005, 0.004
**Lag** **× BMI**	−0.002	0.001, 0.077		
**Lag** **× Medical conditions**	−0.008	0.005, 0.066		
**Lag** **× Purpose in life**	−0.001	0.011, 0.941		

This table shows the results (Estimate, S.E. [Standard Error] and *p*-value) of two linear mixed effect models with total daily physical activity as the outcome. The terms included in the first model are shown on the left and the results for each term are shown in the middle column. This model includes demographic terms age, sex, and education and covariates from the other groups that survived stage 2 and terms for their interaction with time to identify the covariates independently related to the rate of change in total daily physical activity. The second model in the right column includes only demographic terms and the three non-demographic covariates that remained associated with total daily physical activity in the final model.

**Table 4 sensors-23-04152-t004:** Variance of Declining Total Daily Physical Activity Explained by Key Covariates.

Covariates	Percentage of Variance	
**All demographics**	11.6%	
**All non-demographic**	9.4%	
**Individual non-demographic**	Minimum %	Maximum %
**Fractal alpha 2**	0.6%	1.5%
**Motor abilities**	3.8%	8.9%
**IADL disability**	1.8%	6.2%

## Data Availability

All data included in these analyses are available via the Rush Alzheimer’s Disease Center Research Resource Sharing Hub, which can be found at www.radc.rush.edu (accessed on 17 April 2023). It has descriptions of the studies and available data. Any qualified investigator can create an account and submit requests for deidentified data.

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
