# Peer review of "Correlates of Person-Specific Rates of Change in Sensor-Derived Physical Activity Metrics of Daily Living in the Rush Memory and Aging Project"

_sensors, 2023, doi:10.3390/s23084152_

Round 1

Reviewer 1 Report

Dear,

Well-done research, paper, especially the analysis, congratulations. 

Please, see the attachments.

Reviewer 2 Report

This is a very important study attempting to understand the main mental and physical factors that affect the aging population limiting their physical activity.

In general, the paper is well-written and organized. In terms of its length, it is on the longer side. The length may be reduced by shortening the discussion section of the paper.

From the technical perspective, it relies on inferential statistics and regression analysis in the form of mixed effects linear regression. Although the methodology to analyze the regression model is described in Section 2.4, there is no mathematical description of the model or metrics to analyze it involved. Since Sensors is an engineering journal, it would be beneficial to include equations, metrics, and pseudo-descriptions of algorithms.    

Among minor comments I have several.

  1. On page 3, please spell out IQR mentioned in the table and explain how the reader should interpret the data in column 3 of Table 1. Also, at the bottom of page 3 is a set of comments related to covariates provided in the table. Each of the comments ends abruptly. It would be beneficial to provide a more careful description with more to-the-point clarifications.
  2. On page 5, please provide a reference to the z-score or provide a definition.
  3. The text in Figure 1 is barely visible. Please enlarge the font and place the figure in landscape mode.
  4. What is S.E. on pages 7 and 8?
  5. Please label the x-axis in Figure 2. Looking at the left panel in Figure 2, it appears the light gray lines forming the ensemble of lines displayed in the panel have different lengths. Thus, drawing a single predictive line is not the way to treat the data.
  6. On page 10 you conclude that the stage 3 covariates explain only 20 % of the variance in the data. But, is it also possible that the model used in this analysis is not good enough? As an example, a nonlinear version of the mixed effects regression is the mixed effects random forest. Its application could improve results.

Krennmair, P. & Schmid, T. (2022) Flexible domain prediction using mixed effects random forests. Journal of the Royal Statistical Society: Series C (Applied Statistics), 71(5), 1865–1894. Available from: https://doi.org/10.1111/rssc.1260

       7. On page 13, spell out CNS. 

Reviewer 4 Report

The proposed manuscript deals with the investigation of the person specific rates of change of physical activity related to several variables collected at the baseline. Instrumental variables were registered and monitored with a wrist inertial sensor. More than 1000 elderly subjects were enrolled in the study. Several linear mixed-effect models were used to identify covariates associated with the level and annual rate of change of physical activity. The topic and the adopted methodology are in line with current researches. Nevertheless, some revisions must be reported to improve the clarity and the quality of the manuscript. Some important information (in particular in the introduction and discussion) are missing. English form needs to be deeply revised.  Here some suggestions:

-Keywords: I suggest to add some keywords related to the data analysis, for example "linear mixed-effect model", "multivariate modeling". Delete the number after each keyword.

-Introduction: The introduction is poor of content. Only one previous study is described as reference. Additional information of literature researches on the topic must be reported, stressing the main results and limitations. No information about wearable sensors are reported. This section must be deeply revised. Finally, the principal reason and aim of the study must be clearly presented at the end of the introduction and supported by previous literature. 

-Methodology: the several parameters are well described. The statistical analysis is well presented. I have no additional suggestions for this chapter. Well done.

- Results: in Figure 1, I suggest to delete the first table, because it is a repetition of the Table 1. The correlation table can be enlarge for better clarity and readibility. Improve the quality of all the reported figures using vector files.

- Discussion and conclusion: lines 353-366 are not object of discussion and must be reported in the introduction. These sections must be deeply revised to improve the clarity of the contents. In the discussion there are several paragraphs, but they are not linked each other. There are several literature information that must be reported in the introduction section, not in the discussion. Moreover, the authors repeated several times the need to add deeply investiogations, additional parameters, etc... so, it is difficult to understand which are the meaninful results of the present study and how these results can be used/interpreted. There is no a direct discussion of the tables/figures reported in the results sections. Finally, it is not clear if the authors considered to separate the subjects in different groups based on the age and how this choice might influence the results.

Round 2

Reviewer 4 Report

The authors well answer to previous comments and suggestions.
The manuscript has been improved.